# Multi-omic comparative analysis of COVID-19 and bacterial sepsis-induced ARDS

**Richa Batra[1‡], William Whalen[2‡], Sergio Alvarez-Mulett[2], Luis G. Gomez-Escobar[2], Katherine L. Hoffman[3], Will Simmons[3], John Harrington[2], Kelsey Chetnik[1], Mustafa Buyukozkan[1], Elisa Benedetti[1], Mary E. Choi[4], Karsten Suhre[5], Edward Schenck[2], Augustine M. K. Choi[2], Frank Schmidt[6]\*, Soo Jung Cho[2]\*, Jan Krumsiek[1]\***

**1** Department of Physiology and Biophysics, Institute for Computational Biomedicine, Englander Institute for Precision Medicine, Weill Cornell Medicine, New York, New York, United States of America, **2** Department of Medicine, Division of Pulmonary and Critical Care Medicine, Weill Cornell Medicine, New York, New York, United States of America, **3** Department of Population Health Sciences, Division of Biostatistics, Weill Cornell Medicine, New York, New York, United States of America, **4** Division of Nephrology and Hypertension, Joan and Sanford I. Weill Department of Medicine, New York, New York, United States of America, **5** Bioinformatics Core, Weill Cornell Medicine–Qatar, Qatar Foundation, Doha, Qatar, **6** Proteomics Core, Weill Cornell Medicine–Qatar, Qatar Foundation, Doha, Qatar

‡ These authors share first authorship on this work.
\* frs4001@qatar-med.cornell.edu (FS); sjc9006@med.cornell.edu (SJC); jak2043@med.cornell.edu (JK)

**Data Availability Statement:** The data used in this study can be downloaded at https://doi.org/10.6084/m9.figshare.19775359 All R scripts to generate the tables and figures of this paper are

## Abstract

### Background

Acute respiratory distress syndrome (ARDS), a life-threatening condition characterized by hypoxemia and poor lung compliance, is associated with high mortality. ARDS induced by COVID-19 has similar clinical presentations and pathological manifestations as non-COVID-19 ARDS. However, COVID-19 ARDS is associated with a more protracted inflammatory respiratory failure compared to traditional ARDS. Therefore, a comprehensive molecular comparison of ARDS of different etiologies groups may pave the way for more specific clinical interventions.

### Methods and findings

In this study, we compared COVID-19 ARDS (n = 43) and bacterial sepsis-induced (non-COVID-19) ARDS (n = 24) using multi-omic plasma profiles covering 663 metabolites, 1,051 lipids, and 266 proteins. To address both between- and within- ARDS group variabilities we followed two approaches. First, we identified 706 molecules differently abundant between the two ARDS etiologies, revealing more than 40 biological processes differently regulated between the two groups. From these processes, we assembled a cascade of therapeutically relevant pathways downstream of sphingosine metabolism. The analysis suggests a possible overactivation of arginine metabolism involved in long-term sequelae of ARDS and highlights the potential of JAK inhibitors to improve outcomes in bacterial sepsis-induced ARDS. The second part of our study involved the comparison of the two ARDS groups with respect to clinical manifestations. Using a data-driven multi-omic network, we identified signatures of acute kidney injury (AKI) and thrombocytosis within each ARDS group. The AKI-associated network implicated mitochondrial dysregulation which might

available at https://github.com/krumsieklab/covid-ards-plasma.

**Funding:** JK and RB are supported by the National Institute of Aging of the National Institutes of Health under awards 1U19AG063744 and R01AG069901-01. WW was in part supported by NIH T32 HL134629-Martinez. ES is supported by NHLBI K23 HL151876. FS was strongly supported by the Biomedical Research Program at Weill Cornell Medicine in Qatar, a program funded by the Qatar Foundation. JH was funded by the National Institutes of Health grant 5 T32 HL 134629-04. The funders had no role in study design, data collection, and analysis, decision to publish, or preparation of the manuscript.

**Competing interests:** I have read the journal's policy and the authors of this manuscript have the following competing interests: A.M.K.C. is a cofounder and equity stockholder for Proterris, which develops therapeutic uses for carbon monoxide. A.M.K.C. has a use patent on CO. Additionally, A.M.K.C. has a patent in COPD. ES consults for Axle informatics regarding COVID vaccine clinical trials through NIAID. JK holds equity in Chymia LLC and IP in PsyProtix and is cofounder of iollo.

lead to post-ARDS renal-sequalae. The thrombocytosis-associated network hinted at a synergy between prothrombotic processes, namely IL-17, MAPK, TNF signaling pathways, and cell adhesion molecules. Thus, we speculate that combination therapy targeting two or more of these processes may ameliorate thrombocytosis-mediated hypercoagulation.

## Conclusion

We present a first comprehensive molecular characterization of differences between two ARDS etiologies–COVID-19 and bacterial sepsis. Further investigation into the identified pathways will lead to a better understanding of the pathophysiological processes, potentially enabling novel therapeutic interventions.

## Author summary

Acute respiratory distress syndrome (ARDS) is a critical condition of the lung that can arise after severe infections, traumatic injury, or inhalation of toxins. Patients with ARDS are in a complex disease state and at risk of multiple clinical complications, such as thrombosis, lung fibrosis, acute kidney injury, and increased mortality. Currently, there are substantial challenges in the treatment of ARDS due to the high heterogeneity of this condition across patients. Our study compared metabolomic and proteomic changes induced by two different causes of ARDS—COVID-19 infection and bacterial sepsis. We used blood samples of patients from each ARDS group for molecular profiling and identified several hundred molecules from various biological processes differing between the two groups. Based on these results, we made several new propositions: (1) A role of arginine metabolism in long-term sequelae of ARDS. (2) The potential use of JAK-STAT pathway inhibitors for bacterial sepsis-induced ARDS. (3) ARDS-associated mitochondrial dysfunction as a reason for poor prognosis of acute kidney injury that occurred during ARDS. (4) A synergy between prothrombotic processes as a potential reason for hypercoagulation in ARDS. We hypothesize that combination therapy targeting two or more of these prothrombotic processes may ameliorate hypercoagulation.

## 1. Introduction

Acute respiratory distress syndrome (ARDS), a severe form of respiratory failure that is associated with high mortality, emerged as a frequent complication of coronavirus disease 2019 (COVID-19) [1]. ARDS may be induced by other infections (sepsis, influenza), major traumatic injury, or inhalation of toxic chemicals [2]. Clinical presentations and pathological manifestations of COVID-19 ARDS overlap with non-COVID-19 ARDS, including decreased static lung compliance, hypoxemia, hypercarbia, inflammation, thrombosis, and endothelial injury [3–9]. However, COVID-19 ARDS is specifically characterized by a protracted hyperinflammatory state, and may lead to higher rates of thrombosis as well as fibroproliferative lung remodeling [10–12]. These differences in ARDS etiologies have not yet been fully characterized to an extent that would enable timely and tailored clinical care. Moreover, ARDS is a heterogeneous disorder with substantial molecular differences even within a specific ARDS group [13]. Thus, to provide deeper insight into disease pathophysiology and enable etiology-specific

therapeutic interventions, a comprehensive molecular characterization of variations between and within ARDS groups is needed.

Previously, ARDS groups have been studied in comparison to non-ARDS reference groups, such as healthy controls or hospitalized patients without ARDS [5–9, 14–16]. Here, we present the first detailed comparative multi-omic analysis between COVID-19 ARDS (n = 43) and bacterial sepsis-induced (non-COVID-19) ARDS (n = 24). The comprehensive measurement panel included 1,980 molecules, including 663 metabolites, 1,051 lipids, and 266 proteins. We followed a two-step analysis workflow to elucidate the differences between the two ARDS groups. In the first part, we directly compared patients from the two groups to identify differentially abundant molecules. These molecules were involved in various biological processes and may highlight the differences in the pathological manifestation of the groups. Furthermore, we analyzed a set of ARDS-associated biological processes with therapeutic relevance, covering various signaling and metabolic pathways. In the second part of the study, we associated clinical manifestations, including acute kidney injury (AKI), thrombocytosis (platelet count), patient's oxygen in arterial blood to the fraction of the oxygen in the inspired air (PaO2/FiO2), and mortality, with the molecular profiles to identify molecular signatures in each ARDS group. For a systematic comparison of these molecular changes, we performed multi-omic network analysis and identified subnetwork-based signatures. To the best of our knowledge, this is the first study addressing both between and within ARDS group variabilities in a multi-omic setting.

## 2. Results

### 2.1 Cohort characteristics and description of molecular data

We analyzed 67 patients admitted to the intensive care unit at Weill Cornell Medical Center (WCMC)–New York-Presbyterian (NYP) hospital with diagnoses of COVID-19 (n = 43) and bacterial sepsis (n = 24). This cohort included 50 (74.6%) males and 17 (25.4%) females, with a median age of 60. The median mortality rate was 29.9%, with 11 out of 43 in COVID-19 ARDS and 9 out of 24 in bacterial sepsis-induced ARDS. 46.3% of patients suffered from acute kidney injury (AKI), with 16 out of 43 in COVID-19 ARDS and 15 out of 24 in bacterial sepsis-induced ARDS. The sequential organ failure assessment (SOFA) index was comparable between the two groups, with a median of 10 in the COVID-19 group and 9 in the bacterial sepsis group. Further, relatively low plateau pressure and extrinsic PEEP in the bacterial sepsis-induced ARDS group suggest that heterogeneous ventilator strategies were used in this population, which may affect the generalization of some findings from that group. Detailed demographics of the patient cohort are provided in **S1 Table**. Plasma samples from the patients were subjected to untargeted metabolomics, targeted lipidomics, and targeted proteomics profiling. After quality control and preprocessing, 663 metabolites, 1,051 lipids, and 266 proteins were used for further analysis. The overall study and analysis design are shown in **Fig 1**.

### 2.2 Plasma molecular profiles differentiate the two ARDS groups

To assess the molecular differences between COVID-19 and bacterial sepsis-induced ARDS, we analyzed three molecular layers—metabolomic, lipidomic, and proteomic. A total of 175 out of 663 metabolites, 437 out of 1,051 lipids, and 94 out of 266 proteins were differentially abundant between the groups at a 5% false discovery rate (FDR) (**Fig 2A**). Detailed results of this analysis are available in **S2 Table**.

To identify the biological processes underlying the differences between the ARDS groups, the differentially abundant molecules were functionally annotated. Metabolites were annotated

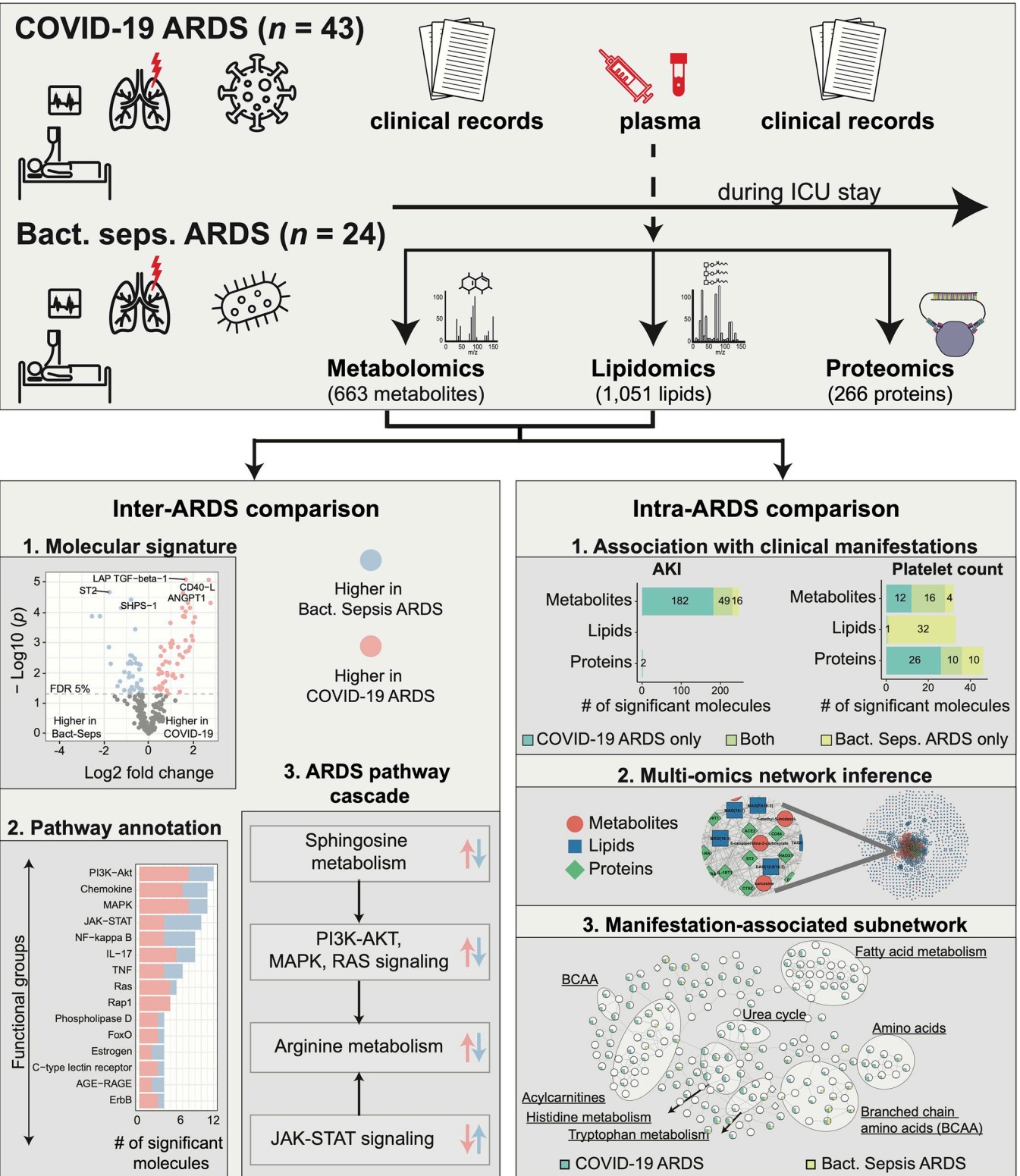

**Fig 1. Study overview.** This study was based on 67 ARDS patients, 43 with COVID-19 and 24 with bacterial sepsis group. Profiling of plasma samples resulted in 1,906 measured molecules, including 663 metabolites, 1,051 lipids, and 266 proteins. For inter-ARDS comparison, we identified molecules and pathways differently regulated between the two ARDS groups. In addition, focusing on several selected pathways with therapeutic relevance, we constructed a cascade of biological processes starting from sphingosine metabolism. For intra-ARDS comparison, we identified molecules associated with clinical manifestations,

including acute kidney injury (AKI), thrombocytosis (platelet count), PaO2/FiO2 ratio, and mortality, within each ARDS group. Further, we constructed a data-driven multi-omic network based on the Gaussian graphical model (GGM). This network was used to generate subnetworks associated with clinical manifestations.

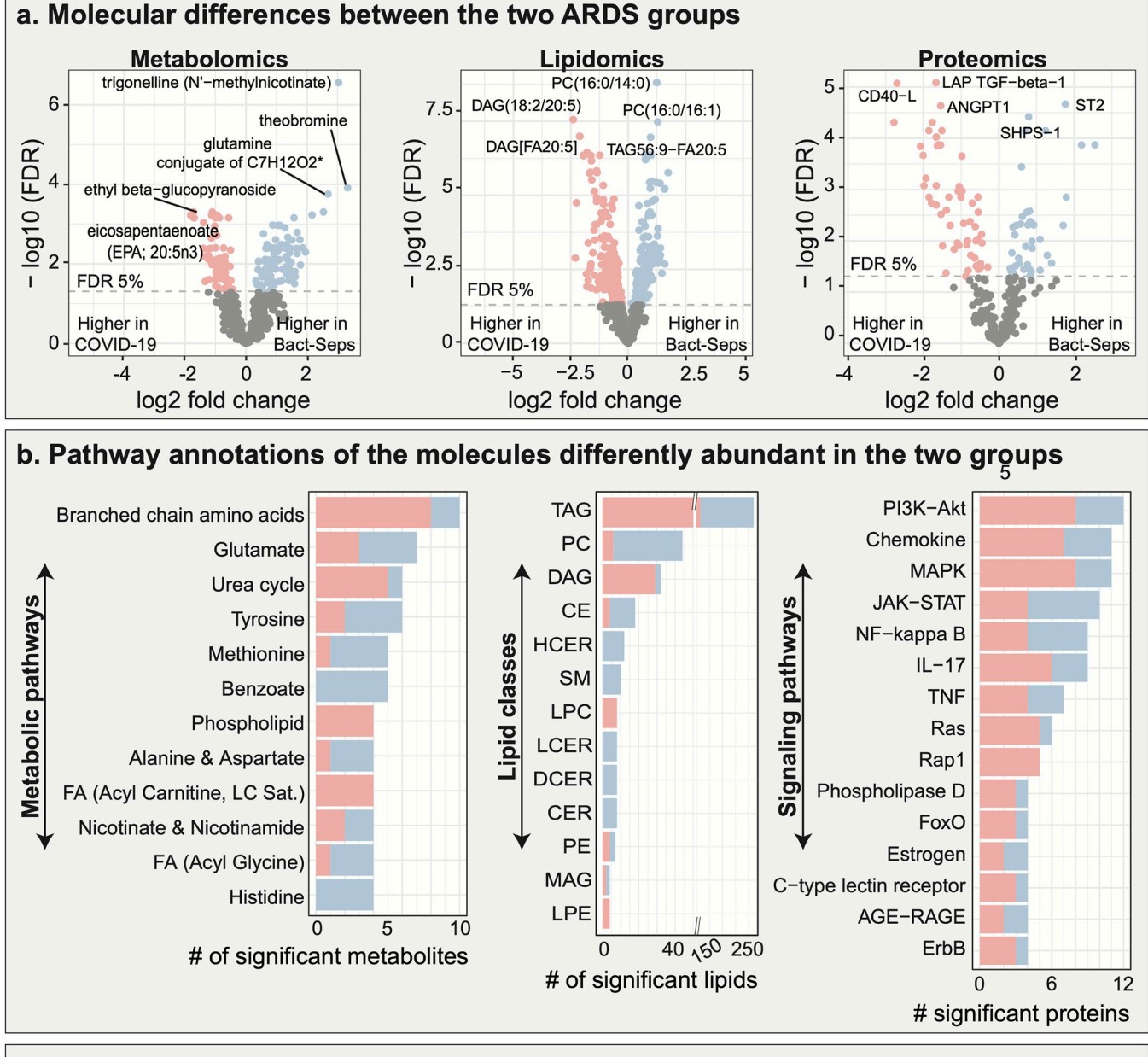

**Fig 2. Multi-omic comparison between COVID-19 ARDS and bacterial sepsis-induced ARDS. a.** Metabolomic, lipidomic, and proteomic analyses between the two ARDS groups. 706 molecules were differently abundant in the two ARDS groups. **b.** Functional annotations of significant molecules. Pathways and classes with at least 4 significant molecules were included in these plots. FDR–false discovery rate. Lipid class abbreviations: TAG–Triacylglycerol, PC–Phosphatidylcholine, DAG–Diacylglycerol, CE–Cholesteryl ester, HCER–Hexosylceramides, Total–total lipids, SM–Sphingomyelin, LPC–Lysophosphatidylcholine, LCER—Lactosylceramides, DCER–Dihydroceramides, CER–Ceramides, PE–Phosphatidylethanolamine, MAG–Monoacylglycerol, LPE—Lysophosphatidylethanolamine, PI—Phosphatidylinositol.

using Metabolon's 'sub-pathways', lipids were annotated by lipid classes, and proteins were annotated using signaling pathways from KEGG [17] (S3 Table). Top ranking pathways are depicted in Fig 2B. Interestingly, several of these pathways from each of the three omics have previously been implicated in COVID-19 ARDS or non-COVID-19 ARDS and will be further discussed per omics and pathway in the following. To corroborate our findings, we used previous studies which have compared these ARDS groups with healthy controls or less severe COVID-19 cases. We followed this route to provide general evidence for the importance of the respective pathway in ARDS, and since parallel studies comparing COVID-19 and non-COVID-19 ARDS at the molecular level in a high-throughput setting were unavailable.

**Metabolic pathways.**   *Branched-chain amino acids (BCAAs)*: In our analysis, 10 metabolites from this pathway were differentially abundant between the ARDS groups. Of these, 8 had higher levels in COVID-19 ARDS compared to bacterial sepsis-induced ARDS, and 2 had lower levels. A previous study based on bronchoalveolar lavage (BAL) fluid reported BCAAs to be higher in non-COVID-19 ARDS groups compared to non-ARDS groups [6]. Further corroborating the role of this metabolite class, BCAAs were recently found to be differently regulated in severe COVID-19 cases compared to mild COVID-19 cases [18]. *Glutamate metabolism*: 7 metabolites from this pathway were found to be differentially abundant between the ARDS groups, of which 4 had higher levels in COVID-19 ARDS compared to bacterial sepsis-induced ARDS and 3 had lower levels. Previous studies have reported elevated glutamate levels in BAL fluid of non-COVID-19 ARDS patients compared to healthy controls, as well as elevated levels of metabolites involved in glutamate metabolism in serum of patients with severe COVID-19 disease compared to healthy controls [18].

**Lipid classes.**   We observed substantial lipidomic changes between the two ARDS groups, with the greatest differences observed in the triacylglycerols (260) and diacylglycerols (32) lipid classes. Triacylglycerols and diacylglycerols levels have been previously associated with mortality in ARDS [19], highlighting the role of lipid metabolism in the prognosis of patients. However, previous studies have reported inconsistent results, with both higher and lower levels of triacylglycerols in the COVID-19 compared to a control group [8, 20]. In our study, 158 TAGs and 29 DAGs had higher levels in COVID-19 ARDS compared to bacterial sepsis-induced ARDS while 102 TAGs and 3 DAGs had lower levels.

**Proteomic pathways.**   *PI3K-AKT signaling*: In our analysis, 12 proteins from the PI3K-AKT pathway were differentially abundant between the ARDS groups, of which 8 molecules had higher levels in COVID-19 ARDS compared to bacterial sepsis-induced ARDS and 4 had lower levels. PI3K-AKT signaling plays a pivotal role in the induction of a hyperinflammatory state [21] and the propagation of acute lung injury [22]. Previous studies found the pathway to be elevated in COVID-19 [23] compared to influenza patients. *MAPK signaling*: 11 proteins from the MAPK pathway were differentially abundant between the ARDS groups, of which 8 proteins had higher levels in COVID-19 ARDS compared to bacterial sepsis-induced ARDS, and 3 had lower levels. The MAPK signaling pathway has previously been reported to promote ARDS [24], and its inhibition has been discussed as a potential therapeutic approach for COVID-19 [25].

Overall, we identified 706 molecules differently abundant between the two ARDS etiologies, revealing more than 40 biological processes (S3 Table) differently regulated between the two groups.

## 2.3. The multi-omic interplay of therapeutically relevant ARDS-associated pathways

Pathway annotations provide functional relevance to the molecules differently abundant between the two ARDS etiologies. However, such an analysis does not provide any insights

into the interplay of these pathways in the context of ARDS pathology. From the list of differently regulated pathways (S3 Table), we selected a few ARDS-associated processes for detailed investigation at the molecular level. We built a cascade of pathways that have been reported to be of pharmaceutical interest in inflammatory or infectious diseases and used literature-based evidence of their interactions (Fig 3). Each of the pathways in this cascade and its therapeutic potential in ARDS is discussed in the following.

**Sphingosine metabolism.** The cascade is built downstream of sphingosine metabolism, involving sphingosine-1 phosphate (S1P) and its receptors (S1PRs) (Fig 3A). In our analysis, sphingosine and sphingosine-1 phosphate levels were higher in COVID-19 compared to bacterial sepsis-induced ARDS.

Sphingosine metabolism plays an important role in immune and vascular systems [26, 27]. S1P and S1PRs have gained considerable attention in the treatment of various inflammatory conditions. For instance, Fingolimod (FTY720), an agonist of S1PR1, is already in clinical use for multiple sclerosis (MS), a chronic autoimmune inflammatory disorder [28]. Moreover, S1P analogs have been studied for the treatment of cytokine storms [29] and pulmonary infections induced by influenza H1N1 and paramyxovirus [30]. Consequently, and owing to the life-threatening hyperinflammatory syndrome induced by SARS-COV2 infections [31], three clinical trials were launched to use S1P-S1RPs agonists (Fingolimod, Opaganib) against COVID-19 (Clinicaltrials.gov identifiers: NCT04280588, NCT04467840, NCT04414618).

**MAPK/ERK, RAS, PI3K-AKT pathways.** S1PRs regulate various downstream signaling pathways, including MAPK/ERK, RAS, PI3K-AKT (Fig 3B) [32], which are involved in viral replication and propagation [33–36]. In our analysis, 12 proteins from the MAPK, PI3K-AKT, and RAS signaling pathways had higher levels in COVID-19 ARDS compared to bacterial sepsis-induced ARDS and 6 proteins had lower levels.

Several components from the MAPK/ERK, RAS, and PI3K-AKT pathways have previously been implicated in COVID-19 and non-COVID-19 ARDS [37]. For instance, PDGF subunits and TGF-beta 1 have been linked to the post-ARDS pulmonary fibrosis [38] and blockage of TNF-R1 by GSK1995057 has been suggested to have therapeutic potential in ARDS [39]. Moreover, pulmonary sequelae of COVID-19 have been associated with IL-6 and TGF-beta via provocation of a fibrotic state [40]. Notably, in our study IL-6 levels were comparable between the two ARDS groups, while its receptor IL-6RA was higher in COVID-19 ARDS compared to bacterial sepsis-induced ARDS.

**Arginine metabolism.** It has been reported that the PI3K-AKT and JAK-STAT signaling pathways induce nitric oxide (NO) production via arginine [41] (Fig 3C). NO production at higher levels mediates lung injury via the formation of toxic oxidants [42]. In our data, arginine levels were higher in COVID-19 ARDS compared to bacterial sepsis-induced ARDS. Notably, NO was not measured in our data.

Arginine depletion strategies that block its conversion to NO and citrulline are effective in inhibiting viral replication (HCV, HIV) [43, 44] and have thus been discussed as a therapeutic approach in the context of the COVID-19 [45]. Furthermore, Karki et. al [46], reported higher levels of the *NOS2* gene (coding for iNOS, one of the enzymes catalyzing nitric oxide production) in severe and critical COVID-19 cases compared to controls [46].

**JAK-STAT signaling pathway.** In our analysis, 4 proteins from the JAK-STAT signaling pathways had higher levels in COVID-19 ARDS compared to bacterial sepsis-induced ARDS, and 6 proteins had lower levels (Fig 3D). Two recent clinical trials have shown improved outcomes in COVID-19 patients that received JAK inhibitors [47, 48].

Taken together, in this section we investigated the interplay of metabolic and signaling events that potentially lead to and propagate the pathophysiology of ARDS. Some of our observations could be corroborated using ARDS-specific literature. The rest of the findings are

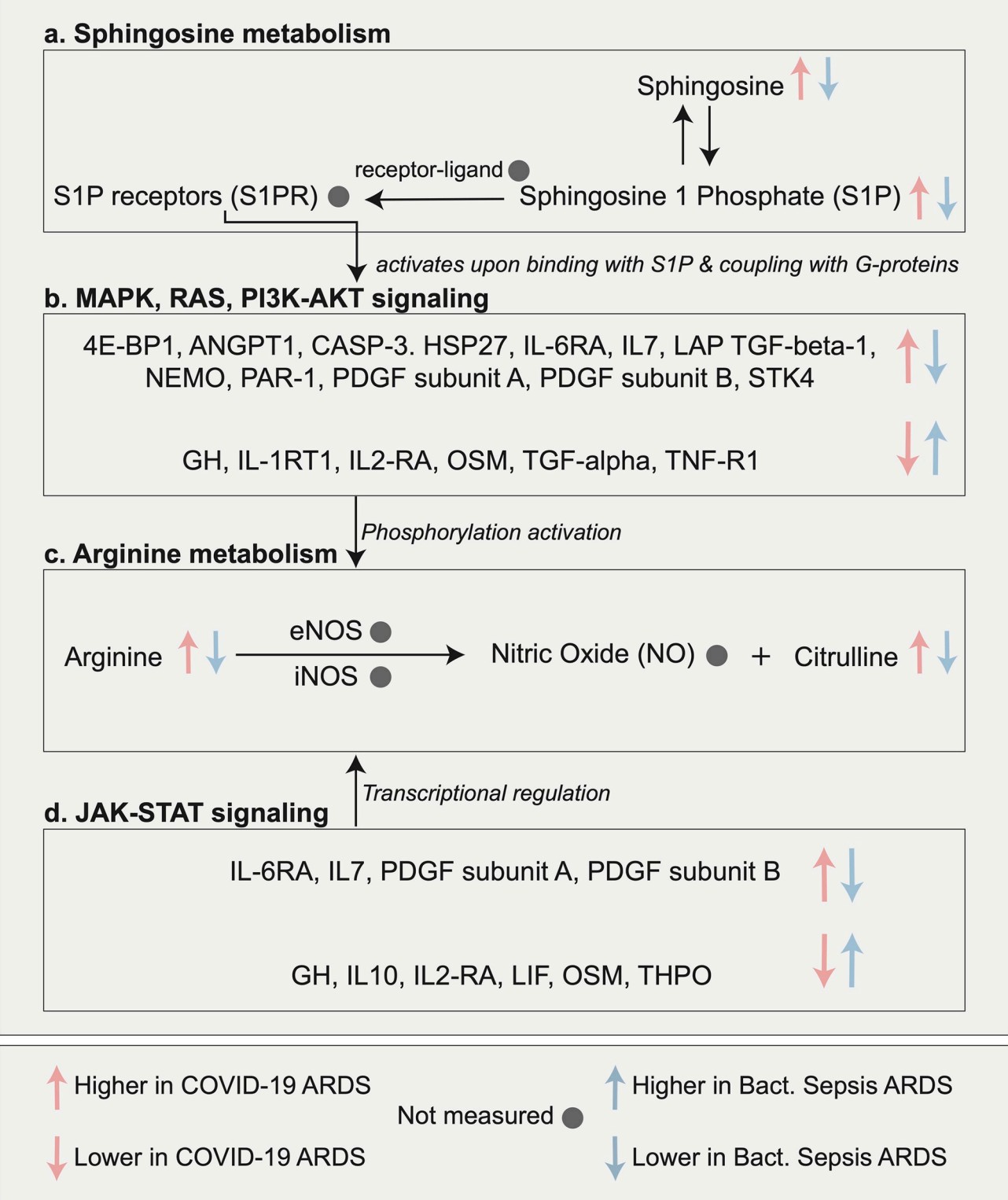

**Fig 3. Multi-omic interplay of selected, therapeutically relevant ARDS-associated pathways.** We selected several pathways that are of pharmaceutical interest in inflammatory or infectious diseases and built the cascade using literature-based knowledge of their interactions. This cascade begins with sphingosine metabolism, which is a target of interest to improve inflammatory conditions [28, 29]. Sphingosine-1 phosphate (S1P) and its receptors (S1PRs) activate MAPK, RAS, and PI3K-AKT signaling pathways downstream [32]. These signaling pathways are central to various biological functions, including viral replication and propagation in host cells [33–36]. Inhibition of these pathways has been discussed as a potential therapeutic approach for

treating ARDS [37, 39]. Further down in this cascade, arginine metabolism can be activated by PI3K-AKT and JAK-STAT signaling pathways [41]. These pathways activate the nitric oxide synthase enzymes, which catalyze the conversion of arginine to nitric oxide. Arginine depletion strategies have been discussed in the context of controlling viral infections [43, 44], and JAK inhibition was found to be effective in improving the outcome of COVID-19 [47, 48].

potentially novel and can be further investigated in a targeted manner to establish a mechanistic understanding of specific pathways/molecules in the context of ARDS. Overall, our analysis suggests that ARDS is an inflammatory process coordinated by multiple cellular processes with severe physiological implications.

## 2.4 Network-based molecular signatures of ARDS-related clinical manifestations

In this part of our study, we compared the differences in omics associations with clinical manifestations across the two ARDS groups. These included acute kidney injury (AKI), thrombocytosis (determined by a pathological increase in platelet count), PaO2/FiO2 ratio (low ratio indicates severe hypoxia), and mortality. For PaO2/FiO2 ratio, we only found significant correlations in the COVID-19 ARDS group and there were no molecules associated with mortality in our data. Thus, these two clinical parameters were not used for the comparison of ARDS groups.

In total, 249 molecules were associated with AKI and 111 molecules associated with platelet count (**Fig 4A**). 76 molecules overlapped between the two ARDS groups, 49 molecules in AKI signature, and 27 molecules in thrombocytosis (platelet count) signature. Detailed results of this analysis are available in **S4**–**S6 Tables**.

To obtain a systematic view of these dysregulated multi-omic molecules across ARDS groups, we adopted a network-based approach. To this end, we generated a data-driven Gaussian graphical model (GGM, **Fig 4B**) [49], which is a partial correlation-based approach to identify interactions between molecules. GGMs have previously been shown to reconstruct biochemical pathways from omics data [50–52] and therefore add biological relevance to results aside from the predefined pathway annotations. We then extracted subnetworks for AKI and thrombocytosis (see methods for details), which will be discussed in the following subsections. An interactive Cytoscape version of the full network and all subnetworks for further exploration are available in **S1 File**.

**2.4.1 Acute kidney injury (AKI) signature.** AKI is a sudden reduction in normal kidney function, leading to an accumulation of toxic waste products in blood. It is a common complication of ARDS, occurring in 44.3% of patients with non-COVID-19 ARDS [53] and 49.5% of patients with COVID-19 ARDS [54]. In our study, 46.3% of patients suffered from AKI, with 16 out of 43 (37.20%) in COVID-19 ARDS and 15 out of 24 (62.50%) in bacterial sepsis-induced ARDS.

Within COVID-19 ARDS patients, 233 molecules were associated with AKI, including 231 metabolites (135 positively, 96 negatively), 2 proteins (both positively), and no lipids. In bacterial sepsis-induced ARDS, 65 metabolites were associated with AKI (46 positively, 19 negatively), and no proteins or lipids. The AKI-associated subnetwork consisted of 245 molecules connected by 373 correlation-based edges. Within this subnetwork, we focused on the largest connected component with 190 molecules and 318 interactions (**Fig 5A**). 121 molecules in this subnetwork were significant in COVID-19 ARDS (5% FDR), and 27 molecules were significant in bacterial sepsis-induced ARDS.

Metabolites from this subnetwork were from two main metabolic groups: amino acid metabolism and fatty acids from the acylcarnitine class. In critical illness, protein catabolism

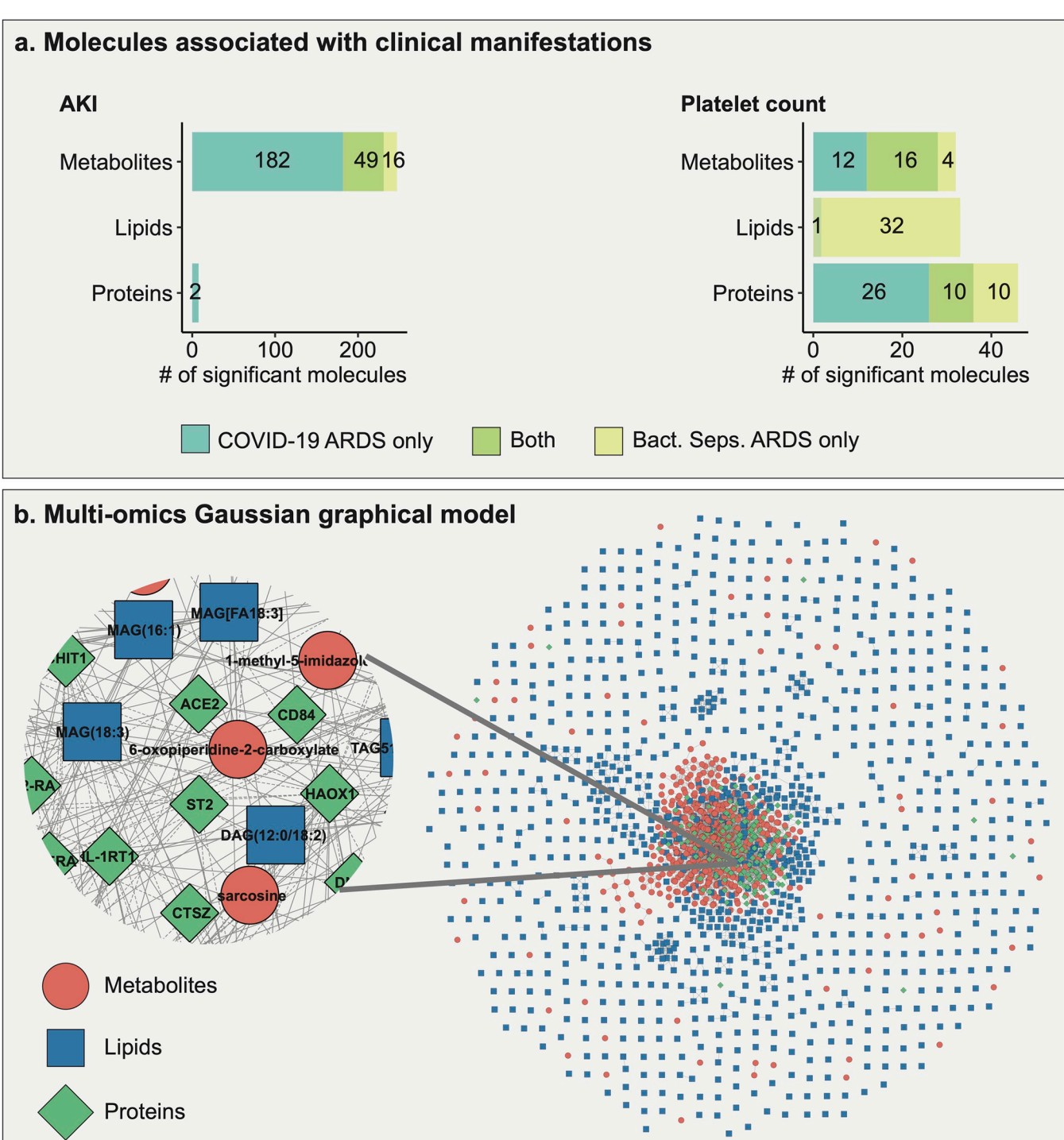

**Fig 4. Overview of intra-ARDS multi-omic signatures within each ARDS. a**. Molecules associated with each of the two clinical manifestations that could be compared across ARDS groups (AKI and platelet count) for each omics layer. 249 molecules were associated with AKI and 111 with platelet count in the ARDS groups. **b**. Gaussian graphical model (GGM) of metabolites, lipids, and proteins. Shapes and colors of the molecules in the network are based on the omics type.

leads to the production of excess amino acids [55, 56], whereas in lung injury, fatty acid oxidation is altered, resulting in the production of acylcarnitines [57]. Amino acid dysregulation has been reported in correlation with COVID-19 severity [58–60], further highlighting the role

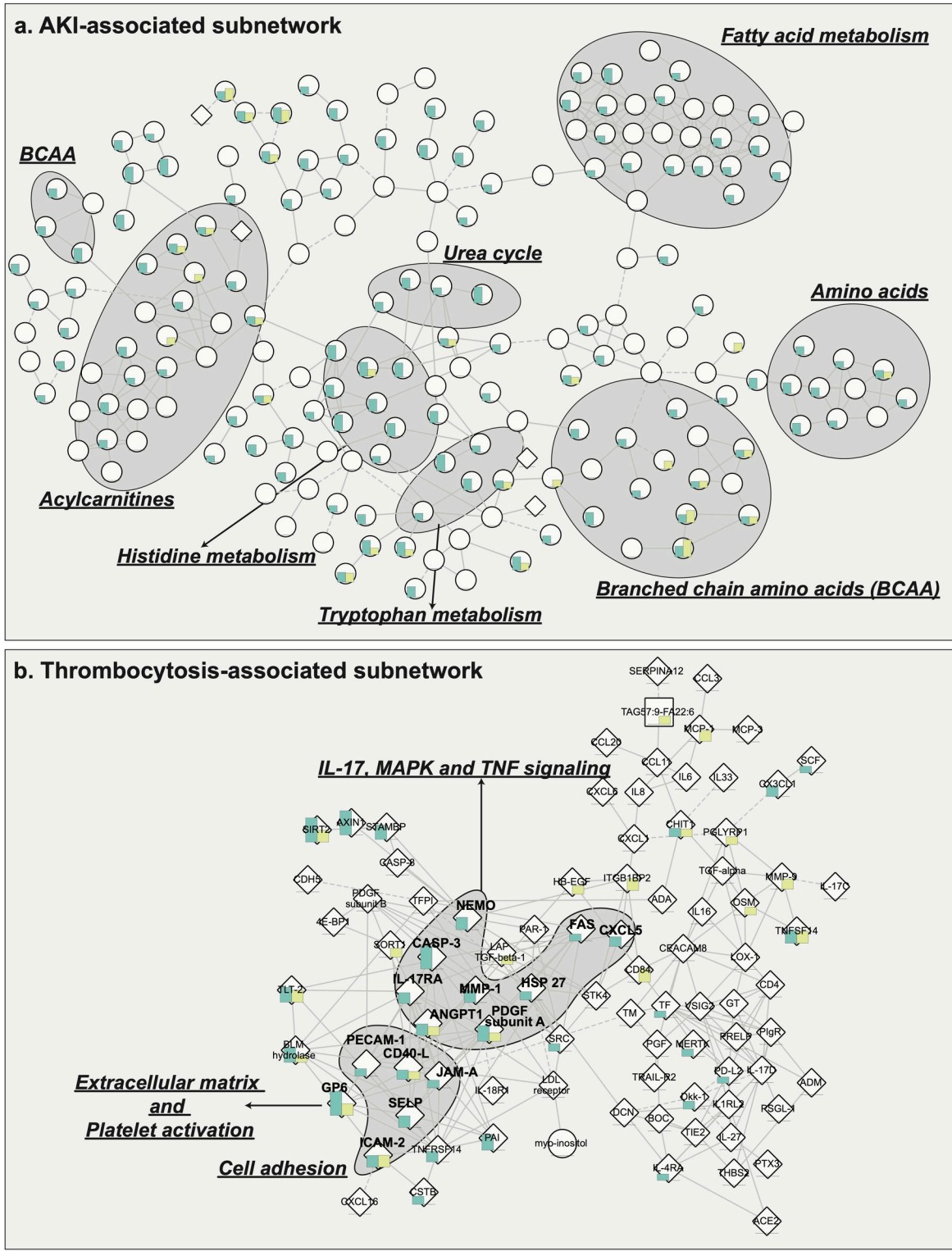

**Fig 5. Subnetwork-based signatures of two clinical manifestations across ARDS groups. a.** AKI-associated subnetwork enriched in amino acids and acylcarnitines, indicating mitochondrial dysfunction. **b**. Thrombocytosis-associated subnetwork enriched in IL-17,

MAPK, TNF signaling pathways, and cell adhesion molecules. These are prothrombotic processes that may lead to hypercoagulative complications in COVID-19.

and relevance of amino acid metabolism. Both amino acids and acylcarnitines are known to be involved in bioenergetic processes mediated by mitochondria [61, 62] and have also been associated with kidney injury [63]. Mitochondrial dysfunction often leads to oxidative stress [64], which is a characteristic feature of COVID-19 as well as ARDS from other etiologies [65, 66]. Therefore, the dysregulation of acylcarnitines and amino acid metabolism observed in plasma suggests widespread mitochondrial dysfunction in AKI associated with ARDS.

**2.4.2 Thrombocytosis signature.** Thrombocytosis is marked by increased production of thrombocytes (platelets), which can be triggered by an underlying condition, such as infection. Platelet activation is crucial for various normal physiological and pathophysiological processes, including hemostasis, thrombosis, and immune response [67]. Thrombosis or coagulopathy is associated with poor prognosis in ARDS patients [11, 68]. Previous studies have reported high incidences of thrombotic complications in COVID-19 ARDS as compared to non-COVID-19 ARDS patients [11]. It has also been postulated that thrombotic manifestation in COVID-19 ARDS is atypical, i.e., despite increased platelet consumption, circulating platelet count is maintained via a compensatory platelet production [11, 68].

Within COVID-19 ARDS patients, 65 molecules were associated with platelet count, including 28 metabolites (24 positively, 4 negatively), 36 proteins (27 positively, 9 negatively), and 1 lipid (positively). In bacterial sepsis-induced ADRS, 73 molecules were associated with platelet count, including 20 metabolites (all positively), 33 lipids (31 positively, 2 negatively), and 20 proteins (19 positively, 1 negatively). The thrombocytosis-associated subnetwork consisted of 165 molecules connected by 268 correlation-based edges. Within this subnetwork, we focused on the largest connected component with 89 molecules and 208 interactions (**Fig 5B**). 33 proteins in the subnetwork were significantly associated with platelet count in COVID-19 ARDS, and 20 proteins were significantly associated with platelet count in bacterial sepsis-induced ARDS.

Proteins in this subnetwork were mainly cell adhesion molecules or belonged to IL-17, MAPK, and TNF signaling pathways. IL-17 signaling is associated with severe inflammatory conditions and works in synergy with TNF signaling to induce vascular genes as well as cell adhesion molecules [69]. After vascular injury, cell adhesion molecules mediate activation of platelet adhesion [70, 71]. IL-17, TNF, and the MAPK signaling pathway have previously been implicated in enabling thrombosis [72–75], a frequent complication in COVID-19 ARDS [76] as well as non-COVID-19 ARDS [77]. Therapies targeting IL-17, TNF, and MAPK have been discussed for COVID-19 treatment [25, 78–81].Taken together, our finding of correlations between thrombocytosis (platelet count) and molecules involved in cell adhesion, IL-17, TNF, MAPK signaling pathways imply a coordinated effort of these pathways toward thrombocytosis-mediated coagulopathy during ARDS.

## 3. Discussion

In this study, we performed a comprehensive multi-omic comparative analysis of COVID-19 and bacterial sepsis-induced ARDS. We profiled plasma samples from 67 patients hospitalized at WCMC/NYP using untargeted metabolomics, untargeted lipidomics, and targeted proteomics profiling, resulting in the quantification of 1,980 molecules. To perform the comparison of molecular differences between these ARDS groups, we followed two approaches. First, to identify differently regulated molecules and biological processes in the two groups, we directly compared the molecular profiles between COVID-19 ARDS and bacterial sepsis-induced

ARDS. Second, to obtain an overview of the similarities and differences in molecular presentation of severity in both groups, we compared molecular associations with clinical manifestation within each group.

For the first part of the study, we identified 706 molecules (metabolites, lipids, proteins) differently abundant between COVID-19 ARDS and bacterial sepsis-induced ARDS. These molecules spanned various biological processes (**Fig 2B**) and may drive the pathological manifestation of the two etiologies. To further contextualize our findings, we built a cascade of ARDS-induced changes in a selected set of interrelated pathways with therapeutic relevance, including sphingosine metabolism, MAPK, RAS, PI3K/AKT signaling, arginine metabolism, and JAK-STAT signaling. This analysis suggested that ARDS is coordinated by multiple cellular processes with severe pathophysiological consequences and led to two main propositions: (1) We speculate that arginine metabolism plays a critical role in the long-term sequelae of ARDS, as arginine metabolism has previously been shown to be altered in the pulmonary fibrosis [82, 83]. (2) We postulate that blockage of JAK-STAT signaling may improve outcomes of bacterial sepsis-induced ARDS. JAK-STAT activation has been implicated in the pathogenesis of ARDS previously [84, 85] and its inhibition has already been shown to improve outcomes of COVID-19 ARDS [86].

For the second part of our study, to examine within-ARDS heterogeneity, we compared molecular profiles within each of the two ARDS groups concerning clinical manifestations. Using a multi-omic network, we identified network-based signatures for AKI and thrombocytosis. The AKI-related subnetwork included deregulated amino acids and acylcarnitines, hinting toward aberrations in bioenergetic processes mediated by mitochondria. Importantly, mitochondrial dysfunction is known to cause the progression of AKI to chronic kidney disease (CKD) [87, 88]. Thus, we hypothesize that mitochondrial dysfunction associated with ARDS may lead to a worse prognosis of AKI. Renal sequelae have been studied in people suffering from severe AKI and requiring renal replacement therapy during COVID-19 infection [40]. The thrombocytosis-related subnetwork included deregulated molecules from IL-17, TNF, MAPK signaling pathways, and cell adhesion molecules. Our findings suggest a synergy between the above-mentioned prothrombotic processes [72, 89, 90] as a likely reason for hypercoagulation in ARDS. We speculate that combination therapy targeting two or more of these processes may ameliorate hypercoagulation.

Our findings are based on a study design with several limitations. (1) Samples in each ARDS group were collected years apart, which may cause variation in the molecular profiles due to differences in sample collection protocols and duration of storage. (2) The number of samples in each ARDS group is limited and imbalanced, with 43 COVID-19 samples and 24 bacterial sepsis samples, which reduces statistical power. (3) Our findings are based on molecules measured in plasma; thus, the measurements may not be representative of the site of ARDS, i.e., the lungs. (4) Metabolomics, lipidomics, and proteomics platforms have limits in terms of the coverage of measured molecules, creating the potential for missed associations. For example, our measurement panel did not contain sphingosine-1 phosphate receptor (S1PR), nitric oxide (NO), and nitric oxide synthases (eNOS, iNOS) from Fig 3; angiopoietin 2 (ANGPT2), which has been associated with COVID-19 ARDS-linked vascular necroptosis [91]; and proinflammatory lipid groups like prostaglandins [92]. Improvements in measurement technology as well as the integration of data from different platforms will help us generate a more complete picture of ARDS-associated molecular changes in the future. (5) Our results are based on statistical associations, and further experiments are needed for validation as well as mechanistic and causal insights.

In summary, we presented a first report on the molecular comparison between two ARDS etiologies: COVID-19 and bacterial sepsis. Our study is a step toward solving two pertinent

clinical challenges associated with ARDS: the identification of novel therapeutic options, and the delineation of heterogeneous pathophysiological manifestations within the ARDS [37]. Even though for COVID-19 ARDS, a few partially effective immunotherapeutic options have been identified in anti-IL-6 therapy and JAK inhibitors, treatment remains a challenge for bacterial sepsis-induced ARDS. Using an inter-ARDS comparison, we highlighted therapeutically relevant signaling and metabolic pathways for ARDS of different etiologies. Using an intra-ARDS analysis, we identified molecular signatures characterizing patient heterogeneity within each ARDS group. Our findings are encouraging and warrant further investigation to evaluate their potential for applicability in clinics and ARDS-specific therapeutic intervention.

## 4. Methods

### 4.1 Ethics statement

The study was approved by the institutional review board at Weill Cornell Medicine (#22–03024534). Written informed consent was received before participation by all patients, except when the institutional review board approved a waiver of informed consent (e.g., for the use of discarded samples and deidentified patient data).

### 4.2 Patient population

The cohort was derived from the Weill Cornell Biobank of Critical Illness (WC-BOCI) at WCMC/NYP. The process for recruitment, data collection, and sample processing has been described previously [93–95]. In brief, the recruits in the WC-BOCI database were patients admitted to the intensive care unit with valid consent between October 2014 to May 2021, including 67 patients with COVID-19 ARDS (n = 43) and bacterial sepsis-induced ARDS (n = 24). Two of the patients included in the COVID-19 ARDS group were positive for bacterial culture in blood within 72 hrs prior to sample collection. These secondary infections were disregarded since the progressive COVID-19 infection was considered the primary cause of ARDS. Clinical data such as demographics, vital signs, labs, and ventilator parameters were obtained through the Weill Cornell-Critical Care Database for Advanced Research (WC-CEDAR) and the Weill Cornell Medicine COVID Institutional Data Repository (COVID-IDR). Additional clinical data was obtained through manual abstraction from the electronic health records.

### 4.3 Clinical manifestations

**Definitions used to diagnose key clinical manifestations used in the study are described below.** *Acute Respiratory Distress Syndrome (ARDS).* We defined ARDS by the Berlin definition [96], which was then adjudicated by two independent pulmonary and critical care attendings after a review of the subject's history, arterial blood gas, and chest X-ray. Bacterial sepsis-induced ARDS was defined if subjects met the criteria for ARDS in addition to meeting the definition for sepsis outlined in The Third International Consensus Definitions for Sepsis and Septic Shock [97]. Subjects were diagnosed with COVID-19 if a viral swab of the nasopharynx tested positive exclusively for SARS-CoV-2 via RT-PCR.

*Acute Kidney Injury (AKI).* AKI was defined based on the 'Kidney Disease: Improving Global Outcomes' definition (KDIGO). KDIGO requires a change of serum creatinine greater than or equal to 0.3 mg/dL within 48 hours, an increase in serum creatinine greater than or equal to 1.5 times baseline serum creatinine known or presumed to have occurred within the past 7 days, or urine output less than or equal to 0.5 mL/kg/hour for six hours [98].

*Sequential Organ Failure Assessment (SOFA)*. The SOFA score is a clinical tool used by clinicians in the ICU to determine the degree of a patient's organ failure. The SOFA score is composed of the following variables, with higher values being assigned for more severe alterations: PaO2/FiO2 (mm Hg), Platelets x $10^3$/μL, Glasgow Coma Scale, Bilirubin (mg/dL), Mean Arterial Pressure or administration of vasoactive required, and Creatinine (mg/dL) [99].

## 4.4 Sample handling

For each participant, whole blood (6–10 mL) was drawn into EDTA-coated blood collection tubes (BD Pharmingen, San Jose, CA). Samples were stored at 4˚C and centrifuged within 4 hours of collection to obtain plasma. Plasma was separated and divided into aliquots and kept at -80˚C. Previous studies have shown that these conditions ensure stable analytes for the metabolomics, lipidomics, and proteomics platforms used in our study [100–102]. Samples were thawed and inactivated in different ways: For the metabolic profiling, x3 sample volume of HPLC grade ethanol was added; for the proteomics analysis, the samples were heat-inactivated in a water bath of 56˚C for 15 minutes. After these processes, the samples were again stored at -80˚C until the omic profiling were performed. Samples were collected after ICU admission. For bacterial sepsis ARDS samples, the median was 1.5 days after admission, with an interquartile range: 1.0–2.0, and for COVID-19 ARDS, the median was 6 days with an interquartile range 3.5–9.5.

## 4.5 Proteomic profiling

This assay was performed using the Olink platform (Uppsala, Sweden) at the Proteomics Core of Weill Cornell Medicine-Qatar, according to the manufacturer's instructions. We used the Inflammation, Cardiovascular II, and Cardiovascular III panels. High throughput real-time PCR of reporter DNA linked to protein-specific antibodies was performed on a 96-well integrated fluidic circuits chip (Fluidigm, San Francisco, CA). Each sample was spiked with quality controls to monitor the incubation, extension, and detection steps of the assay. Additionally, samples representing external, negative, and inter-plate controls were included in each analysis run. From the raw data, real time PCR cycle threshold (Ct) values were extracted using Fluidigm reverse transcription polymerase chain reaction (RT-PCR) analysis software at a quality threshold of 0.5 and linear baseline correction. Ct values were further processed using the Olink NPX manager software (Olink, Uppsala, Sweden). Here, log2-transformed Ct values from each sample and analyte were normalized based on spiked-in extension controls and scale-inverted to obtain Normalized log2-scaled Protein Expression (NPX) values. NPX values were adjusted based on the median of interplate controls (IPC) for each protein and intensity median scaled between all samples and plates.

## 4.6 Metabolomic profiling

This assay was performed by Metabolon, Inc (Morrisville, NC) which utilizes ultrahigh performance liquid chromatograph-tandem mass spectroscopy (UPLC-MS/MS).

Sample preparation was performed using the automated MicroLab STAR system from Hamilton Company. For quality control, before extraction, several recovery standards were added. For extraction, methanol with vigorous shaking followed by centrifugation was used to remove protein, dissociate small molecules bound to protein or trapped in the precipitated protein matrix, and recover chemically diverse metabolites. The resulting extract was placed briefly on a TurboVap® (Zymark) to remove the organic solvent and stored overnight under nitrogen before preparation for analysis.

For quality assurance/quality control (QA/QC), several types of controls were analyzed in concert with the experimental samples that allowed instrument performance monitoring and aided chromatographic alignment. Instrument variability was determined by calculating the median relative standard deviation (RSD) for the standards that were added to each sample before injection into the mass spectrometers. Overall process variability was determined by calculating the median RSD for all endogenous metabolites (i.e., non-instrument standards) present in 100% of the pooled matrix samples. Experimental samples were randomized across the platform run with QC samples spaced evenly among the injections.

For Ultrahigh Performance Liquid Chromatography-Tandem Mass Spectroscopy (UPLC-MS/MS), the sample extract was dried and then reconstituted in solvents compatible with each of the four mass spectroscopic methods. Each reconstitution solvent contained a series of standards at fixed concentrations to ensure injection and chromatographic consistency. The methods were optimized for acidic positive ion hydrophilic compounds, acidic positive ion hydrophobic compounds, and basic negative ions, the fourth aliquot was analyzed via negative ionization.

For metabolite identification, raw data was extracted, peak-identified and QC processed using Metabolon's hardware and software. Metabolon maintains a library based on authenticated standards that contain the retention time/index (RI), mass to charge ratio (*m/z*), and chromatographic data (including MS/MS spectral data) on all molecules present in the library. A variety of curation procedures were carried out to ensure that a high-quality data set was made available for statistical analysis and data interpretation.

## 4.7 Lipidomic profiling

This assay was also performed by Metabolon, Inc.

For sample preparation, lipids were extracted from the biofluid in the presence of deuterated internal standards using an automated BUME extraction according to the method of Lofgren et al. [103].

For MS analysis, the extracts were dried under nitrogen and reconstituted in a dichloromethane:methanol solution containing ammonium acetate. The extracts were transferred to vials for infusion-MS analysis, performed on a Shimadzu LC with nano PEEK tubing and the Sciex SelexIon-5500 QTRAP. The samples were analyzed via both positive and negative mode electrospray. The 5500 QTRAP was operated in MRM mode with a total of more than 1,100 MRMs. Individual lipid species were quantified by taking the ratio of the signal intensity of each target compound to that of its assigned internal standard, then multiplying it by the concentration of internal standard added to the sample. Lipid class concentrations were calculated from the sum of all molecular species within a class, and fatty acid compositions were determined by calculating the proportion of each class comprised of individual fatty acids.

## 4.8 Data preprocessing

Metabolites, lipids, and proteins with more than 25% missing values were removed, leaving 663 out of 1,005 measured metabolites, 1,051 out of 1,218 measured lipids, and 266 out of 276 measured proteins. Sample-wise variation in the data was corrected using probabilistic quotient normalization [104], followed by $\log_2$ transformation. The remaining missing values were imputed using a k-nearest-neighbor-based algorithm [105]. Ten proteins (CCL3, CXCL1, FGF-21, FGF-23, IL-18, IL-6, MCP-1, OPG, SCF, uPA) were measured in multiple Olink panels, and the replicate values for each sample were averaged prior to statistical analysis. All data processing was performed using the maplet R package [106].

To evaluate possible influences of medication on the omics profiles, a linear stepwise backward selection approach was used [107] using medications administered on at least 4 patients. No significant effect of medication was identified in COVID-19 patients (**S7 Table**). This analysis was not performed for the bacterial sepsis-induced ARDS group as medication information was available for only 4 patients.

### 4.9 Differential analysis of molecules

The metabolite, lipid, and protein associations were computed using linear models with the molecules as response variables and diagnosis/clinical manifestations (ARDS group, AKI levels, thrombocyte/platelet count, mortality status) as predictors. Since demographic factors including age, sex, and BMI are considered determinants of COVID-19 severity [108], we did not consider them as covariates in the models. Multiple hypothesis testing was accounted for by correcting the p-values using the Benjamini-Hochberg (BH) method [109]. All of these analyses were performed using the maplet R package [106].

### 4.10 Pathway annotation and filtering

Metabolites were annotated using Metabolon's 'sub-pathway' groups, lipids were annotated by lipid classes, and proteins were annotated using signaling pathways from Kyoto Encyclopedia of Genes and Genomes (KEGG) [17]. The complete list of pathways annotated to the significant molecules is available in **S3 Table**. For Fig 2B, only pathways/classes with at least 4 significant molecules were included. Moreover, only Metabolon's sub-pathways with the term 'metabolism' and only KEGG pathways with the phrase 'signaling pathway' were considered for this analysis.

### 4.11 Multi-Omic network inference

A partial correlation-based Gaussian graphical model (GGM) was computed using the GeneNet R package [110] to infer a multi-omic network. Partial correlations with FDR < 0.05 were used for network construction between molecules. This multi-omic network was annotated with a score which was computed for each molecule/outcome combination as follows:

$p_{score} = -\log_{10}(p.adj) \cdot d$, where $p.adj$ is the adjusted p-value of the model, and $d$ is the direction (-1/1) of the association based on test statistic (positive or negative correlation with the outcome). This score was used to color the nodes in **Fig 5**.

Subnetworks associated with clinical manifestations were generated from the multi-omic network by selecting the molecules significantly associated with the specific clinical manifestation at 5% FDR and at 10% FDR that interact with the molecules significant at 5% FDR. Within this subnetwork, we focused on the largest connected component.

## Supporting information

**S1 Table. Patient demographics and clinical manifestations.**
(XLSX)

**S2 Table. Molecules differently abundant between COVID-19 ARDS and Bacterial sepsis-induced ARDS.**
(XLSX)

**S3 Table. Pathways associated with molecules differently abundant between COVID-19 ARDS and Bacterial sepsis-induced ARDS.**
(XLSX)

**S4 Table. Metabolites associated with clinical manifestations within COVID-19 ARDS and Bacterial sepsis-induced ARDS.**
(XLSX)

**S5 Table. Lipids associated with clinical manifestations COVID-19 ARDS and Bacterial sepsis-induced ARDS.**
(XLSX)

**S6 Table. Proteins associated with clinical manifestations COVID-19 ARDS and within Bacterial sepsis-induced ARDS.**
(XLSX)

**S7 Table. Medication classes and their effects on the omics profiles, metabolomics, lipidomics, and proteomics.**
(XLSX)

**S1 File. An interactive Cytoscape version of the full multi-omics network and all subnetworks used in this study.**
(ZIP)

## Author Contributions

**Conceptualization:** Richa Batra, William Whalen, Augustine M. K. Choi, Soo Jung Cho, Jan Krumsiek.

**Data curation:** Sergio Alvarez-Mulett, Luis G. Gomez-Escobar, Katherine L. Hoffman, Will Simmons.

**Formal analysis:** Richa Batra.

**Funding acquisition:** Augustine M. K. Choi, Jan Krumsiek.

**Investigation:** Richa Batra, William Whalen, John Harrington, Mary E. Choi, Edward Schenck, Soo Jung Cho, Jan Krumsiek.

**Methodology:** Richa Batra, Mustafa Buyukozkan, Elisa Benedetti, Jan Krumsiek.

**Resources:** Karsten Suhre, Augustine M. K. Choi, Frank Schmidt, Jan Krumsiek.

**Software:** Richa Batra, Kelsey Chetnik, Jan Krumsiek.

**Supervision:** Edward Schenck, Soo Jung Cho, Jan Krumsiek.

**Visualization:** Richa Batra.

**Writing – original draft:** Richa Batra, William Whalen, Soo Jung Cho, Jan Krumsiek.

**Writing – review & editing:** Richa Batra, Elisa Benedetti, Edward Schenck, Soo Jung Cho, Jan Krumsiek.

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
