## [Decision Letter · Decision Letter 0]

13 Jun 2022

Dear Dr. Krumsiek,

Thank you very much for submitting your manuscript "Multi-omic comparative analysis of COVID-19 and bacterial sepsis-induced ARDS" for consideration at PLOS Pathogens. As with all papers reviewed by the journal, your manuscript was reviewed by members of the editorial board and by several independent reviewers. The reviewers appreciated the attention to an important topic. Based on the reviews, we are likely to accept this manuscript for publication, providing that you modify the manuscript according to the review recommendations. Especially please address queries concerning the data availability, as the Reviewer #3 specified.

Sincerely,

Jacob S. Yount

Associate Editor

PLOS Pathogens

Alexander Gorbalenya

Section Editor

PLOS Pathogens

Kasturi Haldar

Editor-in-Chief

PLOS Pathogens

orcid.org/0000-0001-5065-158X

Michael Malim

Editor-in-Chief

PLOS Pathogens

orcid.org/0000-0002-7699-2064

Reviewer Comments (if any, and for reference):

Reviewer's Responses to Questions

**Part I - Summary**

Reviewer #1: The study by Batra et.al. compares the multi-omic landscape of COVID-19-induced versus sepsis-induced ARDS. The study is descriptive, and in my opinion is well done with an appropriate acknowledge of limitations. I have few questions to ask for clarification:

Reviewer #2: The authors performed a multi-omic comparative analysis of COVID-19 and bacterial sepsis-induced ARDS. They profiled plasma samples using metabolomics, lipodomics, and proteomics. Firstly, after comparing the molecular profiles between COVID-19 ARDS and bacterial sepsis-induced ARDS, they identified 706 differentially regulated molecules and 40 biological processes. They found that the overactivation of arginine metabolism involved in long-term sequelae of ARDS and that JAK inhibitors may improve outcomes in bacterial sepsis-induced ARDS. Secondly, they compared molecular associations with clinical manifestation to obtain an overview of the similarities and differences in molecular presentation of severity in both groups. They found that mitochondrial dysregulation might lead to post-ARDS renal-sequalae, and there was a synergy between prothrombotic processes, namely IL-17, MAPK, TNF signaling pathways, and cell adhesion molecules.

In general, this is a multi-omics study investigating the molecular characterization of differences between two ARDS etiologies. The interpretation of results and the conclusions are appropriate, and the work is important and can provide useful information. It has the potential for novel therapeutic development. However, there are a few issues to be addressed.

Reviewer #3: The team led by Jan Krumsiek (strong researcher in this field) delivers an interesting multi-omics comparative analysis of COVID-19 and bacterial sepsis-induced ARDS.

(i) They identified 706 molecules differently abundant

between the two ARDS etiologies, revealing more than 40 biological processes differently

regulated between the two groups and assembled a cascade of

therapeutically relevant pathways downstream of sphingosine metabolism.

(ii) The analysis suggests

a possible overactivation of arginine metabolism involved in long-term sequelae of ARDS and

highlights the potential of JAK inhibitors to improve outcomes in bacterial sepsis-induced ARDS.

(iii) The second part of our study involved the comparison of the two ARDS groups with respect to

clinical manifestations. Using a data-driven multi-omic network, we identified signatures of acute

kidney injury (AKI) and thrombocytosis within each ARDS group.

(iv) The AKI-associated network

implicated mitochondrial dysregulation which might lead to post-ARDS renal-sequalae.

(v) The thrombocytosis-associated network hinted at a synergy between prothrombotic processes,

namely IL-17, MAPK, TNF signaling pathways, and cell adhesion molecules.

 A combination therapy targeting two or more of these processes may ameliorate

thrombocytosis-mediated hypercoagulation.

**Part II – Major Issues: Key Experiments Required for Acceptance**

Reviewer #1: 1) COVID-19 ARDS has a SOFA score of 10 and a mortality of 26%, which looks rather low for this type of severity

2) Additionally, the plateau pressure of sepsis ARDS is 20 with a quite low peep, and a very low P/F ratio. There is some disconnect there, can the authors confirm that the classification is correct? What is the ventilatory mode of these patients? Is that uniform? My feeling is that there is heterogeneity of vent settings, which could be a limitation that needs to be addressed as well.

3) When were the blood samples collected? At hospital admission? Later?

4) Can the authors explain the medications patients received at the moment of sampling? That could highly confound the omic results.

5) Was IL-6 measured? Its seems to have been but I cannot find it in the test…How does that compare between both types of ARDS? Some studies found this biomarker elevated in COVID19 (JAMA Intern Med 2020; 180: 1152–54. Intensive Care Med 2020; 46: 846–48) but not all of them (Am J Physiol Regul Integr Comp Physiol

2021; 320: R250–57) Please discuss.

Reviewer #2: 1. Please confirm whether it is serum or plasma. In the section of 4.3 sample handling, it is written as serum, whereas plasma is written elsewhere in the manuscript.

2. The authors stated that samples were stored at 4° C for 1-5 days before being transferred into 80° C freezer. Does this operation affect the expression of metabolites and proteins in the samples? In addition, the samples from COVID-19 ARDS group were collected after 2020, while the earliest samples from bacterial sepsis-induced ARDS group were collected from 2014. What’s the impact of the storage on the expression of metabolites. QC data should be provided.

Reviewer #3: nil

**Part III – Minor Issues: Editorial and Data Presentation Modifications**

Reviewer #1: NA

Reviewer #2: 1. In section 4.8. Differential analysis of molecules, why not to use fold change cutoff to select more robust molecules?

2. The authors did not specify if the COVID-19 patients with sepsis were excluded in the Method section?

3. The last sentence in the section of introduction: strictly speaking, this is not a large-scale study.

4. “AKI is a sudden reduction in normal kidney function leading to an accumulation of toxic waste products in blood” to “AKI is a sudden reduction in normal kidney function, leading to an accumulation of toxic waste products in blood”

Reviewer #3: This study provides interesting and valuable data, but the presentation and the data presentation has to be improved to come here well across.

--1st minor but essential point: Data sharing

There seem to be four data-sets ready for download at

https://doi.org/10.6084/m9.figshare.19775359

This is nice, but could the authors be a bit more explicit in the manuscript what are these data,

and, of highest interest to the reader, where do I find the raw data of the study?

Furthermore, it would be nice to deposit all original data in a public repository.

--2nd minor but essential point: Insights on the data collection processes

some details about the data collection process would be really interesting and should be presented:

Which metabolites were difficult to collect?

1,051 lipids, and 266 proteins is fine, but which groups of lipids (e.g. complex lipids such as inflammation mediators) and of proteins (e.g. membrane proteins) are with this experimental set-up difficult to collect?

b) Really good would be that instead of the easy pathway analysis of the metabolites collected the authors educate the reader which blind spots are there, which metabolites can we not follow with this type of analysis? Which of them are probably relevant in this pathophysiology?

--Essential: Make clear the added value of the study: We have currently this conclusion

"Taken together, our finding of correlations between thrombocytosis (platelet count) and molecules

involved in cell adhesion, IL-17, TNF, MAPK signaling pathways imply a coordinated effort of

these pathways toward thrombocytosis-mediated coagulopathy during ARDS"

Now for this conclusion we do not need this study, this is well known from previous publications on the topic and, even more, we can just do a platelet count to see that we have this dangerous ARDS state. Similarly, clinical diagnosis and standard markers allow to easily distinguish between viral or bacterial etiology of the sepsis (starting e.g. from a lymphocyte count).

So make a bit more clear to the reader what you really gain from your metabolite and proteomics view:

Are there some lipids or proteins visible which were previously not recognized by other groups?

Are there interesting and new clinical implications?

To give an exciting example (but maybe just not possible with your current data): is there an early warning metabolite for thromobocytosis, active long before there is thrombocytosis (would be highly interesting)?

Do we anyway have metabolites singling out risk patients from safe patients?

I think you are for these points already on the right track singling out markers for kidney injury and for ROS damage / mitochondrial damage, but it would be nice to expand on this and as a general point make a bit more the added value of your network view and multiomics strategy clear compared to a good clinician and simple laboratory tests also of course available.

The different networks created and the nice views allowing zoom in and out are of course highly appreciated, but sharpening here the conclusions will enhance this paper even further.

PLOS authors have the option to publish the peer review history of their article (what does this mean?). If published, this will include your full peer review and any attached files.

Reviewer #1: No

Reviewer #2: **Yes: **Tiannan Guo

Reviewer #3: No

Figure Files:

Data Requirements:

Reproducibility:

References:

---

## [Editor Report · Decision Letter 1]

19 Aug 2022

Dear Dr. Krumsiek,

We are pleased to inform you that your manuscript 'Multi-omic comparative analysis of COVID-19 and bacterial sepsis-induced ARDS' has been provisionally accepted for publication in PLOS Pathogens.

Best regards,

Jacob S. Yount

Associate Editor

PLOS Pathogens

Alexander Gorbalenya

Section Editor

PLOS Pathogens

Kasturi Haldar

Editor-in-Chief

PLOS Pathogens

orcid.org/0000-0001-5065-158X

Michael Malim

Editor-in-Chief

PLOS Pathogens

orcid.org/0000-0002-7699-2064
---

## [Editor Report · Acceptance letter]

4 Sep 2022

Dear Dr. Krumsiek,

We are delighted to inform you that your manuscript, "Multi-omic comparative analysis of COVID-19 and bacterial sepsis-induced ARDS," has been formally accepted for publication in PLOS Pathogens.

Best regards,

Kasturi Haldar

Editor-in-Chief

PLOS Pathogens

orcid.org/0000-0001-5065-158X

Michael Malim

Editor-in-Chief

PLOS Pathogens

orcid.org/0000-0002-7699-2064